# PR3 levels are impaired in plasma and PBMCs from Arabs with cardiovascular diseases

Abdelkrim Khadir[1], Dhanya Madhu[1], Sina Kavalakatt[1], Preethi Cherian[1], Monira Alarouj[2], Abdullah Bennakhi[2], Jehad Abubaker[1‡]*, Ali Tiss[1‡]*, Naser Elkum[3‡]*

**1** Biochemistry and Molecular Biology Department, Research Division, Dasman Diabetes Institute, Kuwait City, Kuwait, **2** Medical Division, Dasman Diabetes Institute, Kuwait City, Kuwait, **3** Sidra Medicine, Doha, Qatar

‡ These authors are joint senior authors on this work.
* nelkum@sidra.org (NE); ali.tiss@dasmaninstitute.org (AT); jhad.abubaker@dsamaninstiute.org (JA)

**Data Availability Statement:** All relevant data are within the manuscript and its Supporting Information files.

## Abstract

Cardiovascular disease (CVD) risks persist in patients despite treatment. CVD susceptibility also varies with sex and ethnicity and is not entirely explained by conventional CVD risk factors. The aim of the present study was to identify novel CVD candidate markers in circulating Peripheral blood mononuclear cells (PBMCs) and plasma from Arab obese subjects with and without CVD using proteomic approaches. Human adults with confirmed CVD (n = 208) and matched non-CVD controls (n = 152) living in Kuwait were examined in the present cross-sectional study. Anthropometric and classical biochemical parameters were determined. We employed a shotgun proteomic profiling approach on PBMCs isolated from a subset of the groups (n = 4, each), and differentially expressed proteins selected between the two groups were validated at the mRNA level using RT-PCR (n = 6, each). Plasma levels of selected proteins from the proteomics profiling: Proteinase-3 (PR3), Annexin-A3 (ANX3), Defensin (DEFA1), and Matrix Metalloproteinase-9 (MMP9), were measured in the entire cohort using human enzyme-linked immunosorbent assay kits and were subsequently correlated with various clinical parameters. Out of the 1407 we identified and quantified from the proteomics profiling, 47 proteins were dysregulated with at least twofold change between the two subject groups. Among the differentially expressed proteins, 11 were confirmed at the mRNA levels. CVD influenced the levels of the shortlisted proteins (MMP9, PR3, ANX3, and DEFA1) in the PBMCs and plasma differentially. Despite the decreased levels of both protein and mRNA in PBMCs, PR3 circulating levels increased significantly in patients with CVD and were influenced by neither diabetes nor statin treatment. No significant changes were; however, observed in the DEFA1, MMP9, and ANX3 levels in plasma. Multivariate logistic regression analysis revealed that only PR3 was independently associated with CVD. Our results suggest that the dysregulation of PR3 levels in plasma and PBMCs reflects underlying residual CVD risks even in the treated population. More prospective and larger studies are required to establish the role of PR3 in CVD progression.

**Funding:** This research project (RA-2010-004) was funded by Kuwait Foundation for the Advancement of Sciences.

**Competing interests:** The authors have declared that no competing interests exist.

**Abbreviations:** ACE, angiotensin-converting enzyme; ANX3, Annexin-A3; BMI, Body mass index; BP, blood pressure; CVD, cardiovascular disease; DBP, Diastolic blood pressure; DEFA1, Defensin A1; FBG, fasting blood glucose; HDL, high-density lipoprotein; hsCRP, high sensitivity CRP; LDL, low-density lipoprotein; MAPK, mitogen-activated protein kinase; MMP9, Matrix Metalloproteinase-9; oxLDL, oxidized low-density lipoprotein; PR3, proteinase-3; SBP, Systolic blood pressure; TG, triglycerides; TC, total cholesterol; WC, waist circumference.

## Introduction

Cardiovascular diseases (CVDs) continue to represent a major health burden and cause of death globally [1]. Several factors, such as lifestyle, diet, age, epigenetics, obesity, and diabetes are associated with the onset and development of CVD. The prevention and treatment of CVD are further hampered by increased rates of drug side effects in populations with diabetes, including potential adverse glycemic effects. In addition, in both atherosclerosis and diabetes conditions, immune system imbalance and metabolic stress and their crosstalk play a pivotal role in the initiation and progression of the disease [2]. This dynamic results in a chronic inflammation process involving a complex network of multiple cells and inflammatory and metabolic stress mediators. In addition, ethnicity influences morbidity in CVD [3]. For instance, we have previously reported that despite their apparently healthier status, Arab females are at higher risk of CVD complications due to their higher levels of oxidative stress [4]. Furthermore, populations in Kuwait and other Gulf Cooperation Council countries have high rates of obesity, diabetes, and CVD [5]. The complexity of CVD etiology highlighted by several studies underscores the need for novel biomarkers for stratification and identification, beyond the traditional ones, for the understanding of the independent contribution of diabetes as a risk factor for both CVD improvement and disease management [6].

Blood circulating biomarkers are presently the subject of intense research in the area of omics due to their accessibility and proximity to affected organs (vessels or heart). Previously, proteomic studies analyzed circulating blood cells such as monocytes, platelets, or endothelial cells [7] and have contributed to the better understanding of their role in atherosclerosis in addition to providing novel disease biomarkers. Peripheral blood mononuclear cells (PBMCs) directly participate in the formation of atherosclerotic lesions and they are useful surrogates for studying CVD mechanisms and several studies have reported that PBMC gene expression dynamics are similar to those observed for atherosclerotic plaque in CVDs [8, 9]. Proteomic approaches offer a powerful tool for the detection of differentially expressed proteins involved in molecular processes associated with CVD, in particular, for the development of multi-marker panels with relatively high sensitivity and/or specificity for risk prediction and the improvement of clinical outcomes. However, the application of identified biomarkers in clinical practice remains limited and most studies have not evaluated the reliability of their findings in clinical contexts.

Considering the gaps associated with the practicality of identified biomarkers in numerous previously conducted omics studies, (i) we applied a shotgun proteomic profiling approach on PBMCs isolated from human subjects with CVD and their matched controls to identify differentially expressed proteins between the two groups; (ii) we validated the expression of selected sets of proteins at mRNA levels; (iii) we assessed the circulating levels of four protein targets using ELISA in a large population of CVD-confirmed patients with and without diabetes, along with their matched non-CVD controls and investigated their potential correlation with various clinical parameters.

## Materials and methods

### Study participants and blood analysis

The current study included 208 Arab adults with confirmed CVD-related events (cases) and 152 healthy controls with no reported CVD-related events. Those subjects and their samples were extracted from the Kuwait Diabetes Epidemiology Program (KDEP) that was conducted between June 2011 and August 2012 at the Dasman Diabetes Institute (DDI) to estimate the prevalence of non-communicable diseases among the resident population of Kuwait [10].

Informed written consent was obtained from all subjects before their enrollment. The present study was approved by the Ethical Review Board of Dasman Diabetes Institute (Protocol RA-2010-004) and conducted in accordance with the ethical guidelines of the Declaration of Helsinki [11]. Anthropometric, physical, and blood marker measurements of the subjects were extracted as previously reported [12], and they included body weight, height, waist circumference, and blood pressure (BP). Blood lipid and glucose levels were measured with a Siemens Dimension Chemistry Analyzer (Diamond Diagnostics, Holliston, MA, USA) and fasting blood glucose (FBG) as well as the lipid profile as follows: triglycerides (TG), total cholesterol (TC), low-density lipoprotein (LDL), and high-density lipoprotein (HDL). Haemoglobin A1c (HbA1c) levels were determined with the Variant™ device (Bio-Rad, Hercules, CA, USA). Circulating plasma levels of oxLDL were determined using the human oxLDL ELISA kits (EIAab, Wuhan, China) and hsCRP levels were measured using a high-sensitivity CRP "hsCRP" ELISA kit (Biovendor, Asheville, NC). Assays were performed as per the manufacturers' instructions. PBMCs were separated using the Ficoll–Hypaque density gradient centrifugation method and then, resuspended in freezing media containing 10% dimethyl sulfoxide and stored in liquid nitrogen. Plasma was also separated, aliquoted, and stored at −80˚C until assayed.

## Proteomics analysis

Frozen cell pellets were treated with lysis buffer (2 M Urea, 4% CHAPS) in the presence of protease inhibitor mixture and phosphatase inhibitor for 1 h at room temperature on a rotator. The protein mixtures were then extracted, quantified, and prepared for mass spectrometry (MS) shotgun analysis as previously reported [13]. Briefly, 20 μg of proteins were reduced, alkylated, and digested with trypsin on a strong cationic exchange (BcMag SCX Magnetic bead slurry, Bioclone, San Diego, CA) Eluates were lyophilized in a speed-vac and stored at −20˚C until subjected to MS analysis.

The analysis of peptide digests was performed using liquid chromatography (Easy nanonLC; Proxeon Biosystems, Denmark) coupled with tandem mass spectrometry (MS/MS; LTQ-Orbitrap Velos, Thermo Scientific, Germany) as previously reported [13]. Raw MS data were processed using MaxQuant v1.2.2.5 (Max Planck Institute, Germany) for Label-free proteomics data analysis and searched against the Homo sapiens International Protein Index (IPI version 3.83) database using default settings. The MaxQuant LFQ, the default method for label-free quantification was used for relative quantification. STRING (http://**string**.embl.de) (filter set at 0.7, high confidence) was used to analyze protein–protein interaction networks using proteins exhibiting significant expression differences between the subjects with CVD and the control subjects.

## Quantitative real-time PCR

Total RNA was extracted from frozen PBMCs using AllPrep RNA/Protein Kit (Qiagen, Inc., Valencia, CA). The cDNA was synthesized from total RNA samples using High Capacity cDNA Reverse Transcription Kits (Applied Biosystems, Foster City, CA). The gene expression assays were performed on a Rotor-Disc 100 system using SYBR Green (Qiagen, Inc., Valencia, CA). Relative expression levels were assessed using the ΔΔCT method and GAPDH was used as an internal control for normalization. Primers used for validation are listed in S1 Table.

## Quantification of circulating proteins by ELISA

Circulating levels of Annexin-A3 (ANX3), Defensin (DEFA1), proteinase-3 (PR3), and Matrix Metalloproteinase-9 (MMP9) proteins were assessed in plasma samples from subjects using the ELISA method using the following kits: Human Annexin A3 ELISA kit (DL-ANX3-Hu,

Donglin, Wuxi, China), Human Defensin HNP1-3 and Proteinase 3 (HK317 and HK384, respectively, Hycult Biotech, Uden, The Netherlands), and the Human MMP-9 Platinum ELISA kit (BM2016/2, eBioscience, San Diego, CA, USA). The assays were performed according to the manufacturer's instructions.

### Statistical analysis

All analyses were performed using SAS v9.4 (SAS Institute, Cary, NC). Descriptive statistics are presented as means ± standard deviation for continuous variables or as numbers and percentages for nominal/categorical variables. Student's *t*-test and chi-squared test were used to evaluate differences between continuous and categorical variables, respectively. Spearman's correlation coefficients were estimated to determine associations between levels of circulating markers (ANX3, DEFA1, MMP9, and PR3) and various clinical and metabolic parameters. A logistic regression analysis was performed to estimate odds ratios (ORs) and to examine the predictive effect of each factor on CVD risk. ORs and their 95% confidence intervals (95% CIs) for associated factors were estimated. All statistical assessments were two-sided and were considered significant at $p < 0.05$.

## Results

### Expression profiling and validation in PBMCs

To investigate the potential presence of proteins discriminating between non-diabetic human adults with reported CVD-related events and their matched healthy controls, we extracted whole PBMC proteins from four subjects in each group. Another set of six subjects from each group were used later for expression validation using mRNA and real-time PCR (RT-PCR). Characteristics of the subjects are displayed in S2 Table. Subjects were closely matched with regard to sex, age, and BMI. Nevertheless, patients with CVD had significantly higher levels of high sensitivity C-reactive protein (hsCRP), but lower TC levels, compared with their controls ($p < 0.05$).

Global proteome quantification was performed using label-free MS of total protein extracts. The entire list of identified and quantified proteins (1407 protein groups) from both groups is listed in S3 Table. After filtering the proteomics data using the criteria of at least two unique peptides and proteins identified in at least three out of the four LC-MS/MS runs, 742 proteins, were kept for use in further analyses. Protein ratios between subjects with CVD and the controls were calculated, and p-values were estimated (S3 Table). The global protein distribution between both conditions was visualized via a volcano plot, which revealed that less than 5% of the protein expression was significantly different between the two groups (data not shown). We generated a final set of differentially expressed proteins with CVD/Control ratio of increase or decrease ≥ twofold change and the analysis yielded 47 proteins (Table 1).

Lastly, to elucidate the relationships between the differentially expressed proteins between the CVD cases and their matched controls, we used STRING to generate a network of their molecular interactions (Fig 1) and it clearly revealed that most of the proteins were directly linked to each other. Our screening activity highlighted the dysregulation of proteins belonging to interrelated pathways, which could have implications for the progression of CVD. In addition, the functional enrichment in our protein network was associated largely with the following biological processes: exocytosis, neutrophil degranulation, vesicle-mediated transport, leukocyte activation, and response to stress.

Table 2 lists quantitative RT-PCR data used for the validation of 18 proteins upregulated or downregulated at least twofold in our proteomic screening. In 11 out of the 18 assessed genes, we noted a comparable trend between mRNA and protein expression levels. Indeed, among the upregulated cluster, only *ARHGAP30*, *RPA2*, *AMPHL*, and *CPNE1* mRNA expression

**Table 1. List of differentially expressed proteins from the proteomics profiling.**

| Protein IDs | Uniprot | Gene Names | Protein Descriptions | Mol. Weight [kDa] | Ratio CVD : C | p-value |
|---|---|---|---|---|---|---|
| IPI00219678 | P05198 | EIF2A | Eukaryotic translation initiation factor 2 subunit 1 | 36.1 | >5 | 0.005 |
| IPI00006167 | O15355 | PPM1C | Protein phosphatase 1G | 59.3 | >5 | 0.015 |
| IPI00186966 | O00499-1 | AMPHL | Isoform IIA of Myc box-dependent-interacting protein 1 | 64.7 | >5 | 0.005 |
| IPI00025721 | Q9UNS2 | COPS3 | COP9 signalosome complex subunit 3 | 47.9 | >5 | 0.004 |
| IPI00012837 | P33176 | KIF5B | Kinesin-1 heavy chain | 109.7 | >5 | 0.001 |
| IPI00744015 | Q13409-1 | DNCI2 | Isoform 2A of Cytoplasmic dynein 1 intermediate chain 2 | 71.5 | >5 | 0.004 |
| IPI00830067 | Q7Z6I6-1 | ARHGAP30 | Isoform 1 of Rho GTPase-activating protein 30 | 118.6 | >5 | 0.024 |
| IPI00646689 | Q9BRA2 | TXNDC17 | Thioredoxin domain-containing protein 17 | 13.9 | >5 | 0.003 |
| IPI00401264 | Q9BS26 | ERP44 | Endoplasmic reticulum resident protein ERp44 | 47.0 | >5 | 0.020 |
| IPI00397701 | B4DP32 | RPS16 | 40S ribosomal protein S16 | 16.4 | >5 | 0.001 |
| IPI00646500 | P15927-3 | REPA2 | Isoform 3 of Replication protein A 32 kDa subunit | 38.8 | >5 | 0.003 |
| IPI00221224 | P15144 | ANPEP | Aminopeptidase N | 109.5 | >5 | 0.034 |
| IPI00909773 | B4DSZ4 | UBCE7 | Ubiquitin carrier protein | 23.9 | >5 | ≤0.001 |
| IPI00021766 | Q9NQC3-1 | RTN4 | Isoform 1 of Reticulon-4 | 129.9 | >5 | 0.006 |
| IPI00027834 | P14866 | HNRNPL | Heterogeneous nuclear ribonucleoprotein L | 64.1 | >5 | 0.002 |
| IPI00220834 | P13010 | XRCC5 | ATP-dependent DNA helicase 2 subunit 2 | 82.7 | **2.14** | 0.009 |
| IPI00296337 | P78527-1 | HYRC | Isoform 1 of DNA-dependent protein kinase catalytic subunit | 469.1 | **2.10** | 0.029 |
| IPI00789159 | A8MUS3 | RPL23A | Ribosomal protein L23a, isoform CRA_a | 21.9 | **2.08** | 0.027 |
| IPI00894409 | B0QZ18 | CPNE1 | copine I isoform b | 59.7 | **2.08** | 0.032 |
| IPI00019563 | B4DWA5 | GIMAP4 | GTPase IMAP family member 4 | 39.0 | **2.04** | 0.054 |
| IPI00017526 | P25815 | S100E | Protein S100-P | 10.4 | **0.50** | 0.034 |
| IPI00292532 | P49913 | CAMP | Cathelicidin antimicrobial peptide precursor | 19.6 | **0.48** | 0.042 |
| IPI00791534 | P02730 | AE1 | 104 kDa protein; Band 3 anion transport protein | 103.9 | **0.46** | 0.020 |
| IPI00478231 | P61586 | ARH12 | Transforming protein RhoA | 21.8 | **0.43** | 0.035 |
| IPI00298860 | P02788 | LTF | cDNA FLJ78440, highly similar to Human lactoferrin | 78.4 | **0.40** | 0.001 |
| IPI00827847 | P17213 | BPI | Bactericidal permeability-increasing protein | 53.9 | **0.38** | 0.037 |
| IPI00410714 | P69905 | HBA1 | Hemoglobin subunit alpha | 15.3 | **0.38** | 0.002 |
| IPI00643623 | A6NII8 | NGAL | Neutrophil gelatinase-associated lipocalin | 22.8 | **0.37** | 0.006 |
| IPI00641737 | P00738 | HP | Haptoglobin isoform 2 | 46.7 | **0.35** | 0.003 |
| IPI00027409 | P24158 | MBN; PRTN3; PR3 | Myeloblastin | 27.8 | **0.34** | ≤0.001 |
| IPI00654755 | P68871 | HBB | Hemoglobin subunit beta | 16.0 | **0.28** | 0.001 |
| IPI00005721 | P59665 | DEFA1 | Neutrophil defensin 1 | 10.2 | **0.26** | ≤0.001 |
| IPI00027846 | P22894 | MMP8 | Neutrophil collagenase | 53.4 | **0.25** | 0.040 |
| IPI00027509 | P14780 | MMP9 | Matrix metalloproteinase-9 | 78.5 | **0.24** | ≤0.001 |
| IPI00473011 | P02042 | HBD | Hemoglobin subunit delta | 16.1 | **0.23** | ≤0.001 |
| IPI00745868 | A6NLK4 | ANXA3 | Annexin A3 | 36.4 | **0.19** | 0.001 |
| IPI00021841 | P02647 | APOA1 | Apolipoprotein A-I; Apolipoprotein A1 | 30.8 | <0.10 | 0.006 |
| IPI00914949 | Q59FP5 | SPTB | Spectrin, beta | 268.2 | <0.10 | 0.002 |
| IPI00893949 | A0N0E2 | HLA-A | Major histocompatibility complex, class I, A | 41.4 | <0.10 | 0.003 |
| IPI00220741 | P02549-1 | SPTA | Isoform 1 of Spectrin alpha chain, erythrocyte | 280.0 | <0.10 | 0.020 |
| IPI00024282 | Q92930 | RAB8B | Ras-related protein Rab-8B | 23.6 | <0.10 | 0.013 |
| IPI00795979 | Q14254 | ESA1 | Flotillin-2 | 53.1 | <0.10 | 0.006 |
| IPI00026516 | P55809 | OXCT | Succinyl-CoA:3-ketoacid-coenzyme A transferase 1 | 56.2 | <0.10 | 0.001 |
| IPI00016255 | Q6P4A8 | PLBD1 | Putative phospholipase B-like 1 | 63.3 | <0.10 | 0.018 |
| IPI00006608 | P05067-1 | A4 | Isoform APP770 of Amyloid beta A4 protein (Fragment) | 86.9 | <0.10 | 0.002 |
| IPI00855785 | P02751-15 | FN | Isoform 15 of Fibronectin | 272.3 | <0.10 | 0.005 |
| IPI00472825 | P10314 | HLAA | HLA class I histocompatibility antigen, A-32 alpha chain | 41.0 | <0.10 | 0.009 |

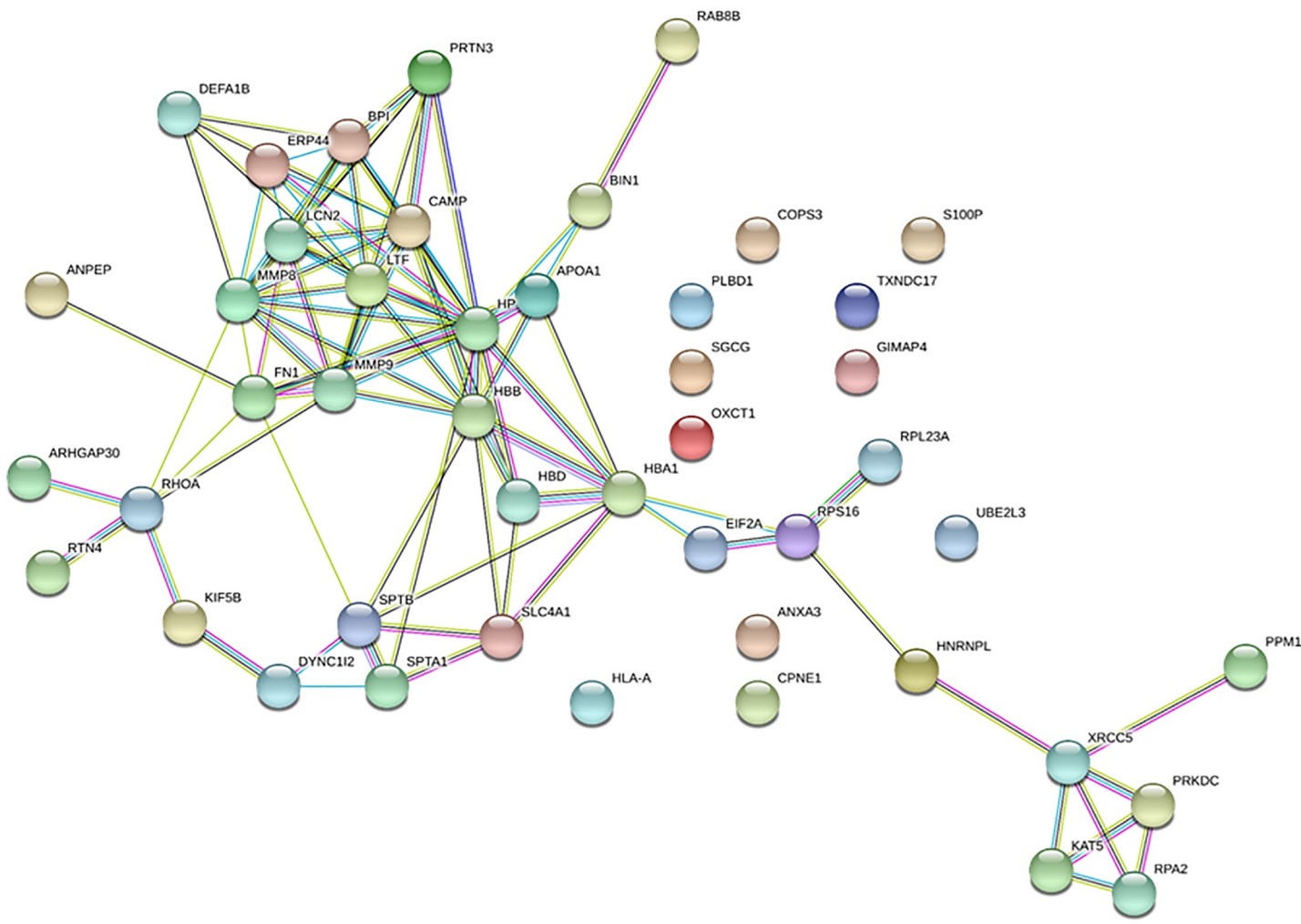

**Fig 1. Relationship network analysis.** Molecular Interaction network of the proteins differentially expressed in the PBMCs from CVD and their controls individuals, was constructed using STRING software and showed a direct connection between most of the proteins affected by CVD. Network nodes represent proteins, edges represent protein-protein associations and line thickness indicates the strength of data support. Color codes are for interactions based on: curated databases (turquoise), experimentally determined (purple), gene neighbourhood (green), gene fusions (red), and gene co-occurrence (blue).

were consistent with the proteomics data, while for the downregulated cluster, the mRNA levels of *EIF2A*, *PR3*, *DEFA1*, *MMP9*, *ANX3*, *SPTA*, and *A4* were consistent with proteomic analysis results.

## Assessment of selected biomarkers in the plasma of subjects with CVD

On the basis of the list of dysregulated proteins and their validation at the mRNA level, we selected four proteins (ANX3, DEFA1, MMP9, and PR3) known to be present in the blood and we assessed their circulating amounts in a large cohort of subjects with reported CVD-related events (n = 208) as well in a matched control group (n = 152). The anthropometric features and metabolic measurements of the study population are listed in Table 3. Both groups were obese (BMI > 30 kg/m$^2$) and had comparable ages. We observed significantly higher levels of glycemia markers (FBG and HbA1c) in the CVD cases while the controls had significantly higher levels of HDL and oxidized low-density lipoprotein (oxLDL) (p < 0.05). With respect to the non-classical markers identified in our study using proteomics, only PR3

**Table 2. Validation of proteomic data with RT-PCR.**

| Gene | Expression ratio CVD / C | |
|---|---|---|
| | Proteomics | RT-PCR |
| **ARHGAP30** | **>5** | **1.7** |
| **RPA2** | **>5** | **1.4** |
| **AMPHL** | **>5** | **1.53** |
| ERP44 | >5 | 0.96 |
| PPMC | >5 | 0.78 |
| TXDNC17 | >5 | 0.65 |
| KIF5B | >5 | 0.73 |
| **CPNE1** | **2.08** | **1.8** |
| **EIF2A** | **0.5** | **0.65** |
| **PR3; PRTN3; MBN** | **0.34** | **0.47** |
| **DEFA1** | **0.26** | **0.65** |
| **MMP9; CLG4B** | **0.24** | **0.7** |
| ANX3 | **0.19** | **0.11** |
| OXCT | <0.1 | 1.4 |
| SPTB | <0.1 | 3 |
| **FLOT2** | **<0.1** | 1.4 |
| **SPTA** | **<0.1** | **0.5** |
| **A4** | **<0.1** | **0.65** |

Bold indicates similar trend between protein and RNA expression levels.

exhibited significantly higher levels in CVD cases (p = 0.0004). Despite the smoking rate being comparable between both groups, there was a higher proportion of subjects with diabetes in the CVD group (p < 0.0001). In addition, most CVD cases had experienced a vascular complication that resulted in coronary or peripheral artery disease (68%) compared with 22% diagnosed with heart failure and 10% with a stroke. Finally, in the CVD group, more than 50% of the individuals were treated with statins and blood pressure-lowering medications while only around a quarter of the control individuals were using such medication.

## Correlation analysis of biomarkers

We used Spearman's rank test to assess correlations between circulating levels of ANX3, DEFA1, MMP9, and PR3 proteins with classical physical and clinical parameters of the participating subjects and the results are displayed in Table 4. Globally, more correlations were obtained using MMP9 compared with ANX3, DEFA1, and PR3, using all subjects or by analyzing the CVD and control groups separately. When all subjects were included, MMP9 was correlated positively with adiposity markers (weight, waist circumference, and BMI), BP, and TG; however, it was inversely correlated with HDL and oxLDL. PR3 and DEFA1 were mainly correlated with BMI, while ANX3 was negatively correlated with BMI. Notably, MMP9 was strongly correlated with ANX3 and DEFA1 levels, both in all subjects as well as in the CVD and control groups separately.

## Trend analysis of biomarkers in the study population based on diabetes and statin treatment

To investigate the influence of medication and diabetes on PR3, ANX3, MMP9, and DEFA1 levels, we segregated our subjects based on either diabetes status (Table 5) or statin treatment

**Table 3. Study population characteristics.**

| Parameter | No CVD (n = 152) | Reported CVD (n = 208) | P-value |
|---|---|---|---|
| *Anthropometric and Metabolic Measurements* | | | |
| Age, years | 52.3 ± 11.1 | 54.45 ± 11.7 | 0.0766 |
| Sex (M/F) | 85/67 | 132/76 | 0.1487 |
| Waist, cm | 101.8 ± 12.7 | 103.3 ± 13.8 | 0.3089 |
| Hip, cm | 109.9 ± 11.9 | 109.3 ± 12.0 | 0.6366 |
| BMI (kg/m$^2$) | 31.3 ± 5.63 | 31.40 ± 5.96 | 0.8625 |
| SBP (mmHg) | 134.9 ± 19.9 | 136.9 ± 20.1 | 0.3596 |
| DBP (mmHg) | 80.1 ± 11.8 | 77.8 ± 12.07 | 0.0753 |
| FBG (mmol/l) | 6.40 ± 2.58 | 7.34 ± 3.47 | **0.0035** |
| HbA1c | 6.70 ± 1.78 | 7.29 ± 2.12 | **0.0007** |
| TC (mmol/l) | 5.29 ± 1.03 | 4.87 ± 1.29 | 0.4882 |
| TG (mmol/l) | 1.67 ± 1.00 | 1.76 ± 1.42 | 0.0884 |
| HDL (mmol/l) | 1.16 ± 0.37 | 1.10 ± 0.35 | **0.0001** |
| LDL (mmol/l) | 3.37 ± 0.88 | 2.97 ± 1.06 | **0.0517** |
| hsCRP (ug/l) | 2.41 (0.20–13.76) | 3.12 (0.09–11.85) | 0.1158 |
| oxLDL (ug/l) | 18.2 (1.32–69.78) | 16.0 (0.03–61.21) | **0.0410** |
| MMP9 | 92.21 (0.20–438.23) | 79.44 (0.51–998.14) | 0.3771 |
| PR3 | 61.60 (14.05–571.30) | 105.54 (5.35–705.45) | **0.0004** |
| Defensin | 6.01 (5.63–115.65) | 6.03 (5.64–83.62) | 0.3176 |
| Annexin A3 | 2.17 (0.66–8.39) | 2.01 (0.56–21.45) | 0.2707 |
| *Risk factors* | | | |
| Smoking | 45 (29.6%) | 51 (24.5%) | 0.2811 |
| Diabetes | 71 (46.7%) | 140 (67.3%) | $<$**0.0001** |
| *Major type of CVD* | | | |
| Coronary and Peripheral Artery Disease | - | 68% | - |
| Heart Failure | - | 22% | - |
| Stroke | - | 10% | - |
| *Medication* | | | |
| Statins | 40 (26%) | 111 (53%) | - |
| Blood pressure treatment | 42 (27.6%) | 117 (56.3%) | - |
| Anti-platelets | - | 27 (13%) | - |

(Table 6). Unexpectedly, TC, LDL, and oxLDL levels were significantly lower both in the non-diabetic cases compared with non-diabetic control and in the diabetic cases when compared with non-diabetic cases subjects. hsCRP, an indicator of inflammation, was higher in the non-diabetic cases compared with the control diabetic cases; however, despite the increased levels in cases with diabetes, the increase was not statistically significant when compared with the levels in the control subjects with diabetes. In addition, while there were no significant differences among the four groups with regard to the levels of ANX3, DEFA1, and MMP9, PR3 levels were significantly higher in cases either with or without diabetes, when compared with their respective control groups.

Statin treatment influences lipid profiles and inflammatory responses. Similarly, we observed abnormal lipid profiles (Table 3) when we compared the lipid profiles between CVD cases and their matched controls, particularly with regard to TC and LDL, which were unexpectedly lower in the CVD case group. When segregating subjects based on statin treatment, the results (Table 6) further revealed decreased levels of TC and LDL in cases treated with statin compared with cases not treated with statin. More importantly, PR3 levels seemed not to

**Table 4. Spearman correlations.**

| Parameter | MMP9 | | | PR3 | | | Annexin A3 | | | Defensin | | |
|---|---|---|---|---|---|---|---|---|---|---|---|---|
| | All | CVD | No CVD | All | CVD | No CVD | All | CVD | No CVD | All | CVD | No CVD |
| Age, years | -0.086 | -0.012 | -0.089 | -0.099 | 0.016 | -0.038 | 0.040 | -0.033 | -0.092 | -0.048 | -0.053 | -0.096 |
| Weight, kg | **0.162**\*\* | 0.142 | 0.102 | 0.034 | 0.068 | 0.056 | -0.008 | **0.232**\*\* | 0.172 | 0.089 | 0.078 | 0.109 |
| Waist, cm | **0.150**\*\* | 0.071 | 0.114 | 0.072 | 0.050 | 0.023 | 0.015 | **0.236**\*\* | 0.099 | 0.012 | 0.022 | 0.129 |
| Hip, cm | 0.093 | 0.081 | 0.088 | **0.127**\* | 0.057 | 0.026 | -0.086 | 0.138 | 0.063 | 0.074 | 0.063 | 0.074 |
| BMI | **0.112**\* | 0.078 | 0.070 | **0.117**\* | 0.075 | 0.025 | **-0.138**\* | **0.189**\* | 0.137 | **0.123**\* | 0.091 | 0.078 |
| systolic | **0.138**\* | -0.054 | 0.091 | -0.039 | 0.053 | -0.124 | **0.111**\* | -0.038 | 0.073 | 0.044 | 0.048 | -0.008 |
| Diastolic | **0.189**\*\*\* | -0.048 | 0.091 | 0.046 | 0.046 | **-0.218**\* | 0.014 | -0.043 | 0.049 | 0.070 | 0.033 | -0.060 |
| FBG | 0.025 | **-0.182**\* | -0.082 | -0.010 | -0.127 | -0.047 | 0.034 | 0.033 | 0.021 | -0.045 | -0.066 | -0.095 |
| HbA1c, % | -0.005 | -0.021 | **-0.223**\*\* | -0.014 | -0.126 | 0.018 | 0.006 | 0.053 | -0.163 | -0.039 | -0.033 | -0.038 |
| TC | 0.072 | -0.037 | 0.022 | -0.077 | **0.225**\*\* | -0.128 | 0.004 | -0.101 | -0.096 | -0.029 | 0.080 | **-0.165**\* |
| TG | **0.218**\*\*\* | -0.149 | 0.117 | -0.001 | 0.113 | 0.039 | **0.138**\* | 0.051 | -0.079 | 0.023 | 0.013 | 0.039 |
| HDL | **-0.240**\*\*\* | 0.026 | -0.124 | -0.033 | 0.045 | -0.047 | **-0.206**\*\* | -0.088 | -0.008 | -0.027 | -0.018 | -0.111 |
| LDL | 0.085 | -0.007 | 0.016 | -0.055 | **0.210**\* | -0.139 | -0.010 | -0.080 | -0.076 | -0.040 | 0.098 | -0.148 |
| Oxldl | **-0.231**\*\*\* | -0.141 | **-0.364**\*\*\* | 0.022 | -0.012 | 0.013 | -0.053 | -0.005 | -0.193 | 0.039 | 0.041 | |
| MMP9 | - | - | - | -0.027 | 0.082 | -0.046 | **0.265**\*\*\* | **0.181**\* | **0.345**\*\*\* | **0.299**\*\* | **0.270**\*\*\* | **0.359**\*\*\* |
| PR3 | -0.028 | 0.023 | -0.036 | - | - | - | -0.079 | 0.005 | -0.036 | 0.099 | **0.184**\* | -0.037 |
| Annexin A3 | **0.265**\*\*\* | **0.181**\* | **0.353**\*\*\* | -0.079 | 0.019 | -0.095 | - | - | - | 0.066 | 0.060 | 0.078 |
| Defensin | **0.298**\*\*\* | **0.271**\*\* | **0.359**\*\*\* | 0.099 | **0.242**\*\* | -0.049 | 0.065 | 0.060 | 0.125 | - | - | - |

be affected by statin treatment as they maintained higher levels both in cases with or cases without statin treatment, compared with their respective control groups.

## Multivariate logistic regression analysis of independent CVD risk predictors

Table 7 presents multiple logistic regression models for CVD in relation to circulating markers adjusted for age, sex, BMI, diabetes, TC, and hypertension. Models 1 to 6 were fitted separately for each biomarker to investigate if they could predict CVD independently. Based on the results of the models, only PR3 and hsCRP could significantly predict CVD (95% CI: 1.211–2.339 and 1.039–1.804; p-value: 0.0019 and 0.0254, respectively). Model 7 presents a multiple logistic regression that includes all the biomarkers in models 1–6 (ANX3, DEFA1, MMP9, PR3, oxLDL, and hsCRP). According to the results of the present analysis showed, subjects with the highest levels of PR3 were more likely to have CVD (OR = 1.683, 95% CI 1.156–2.450).

## Discussion

Despite the value of traditional risk factors and the relatively large panel of biomarkers used to devise a variety of classical treatments for patients with CVD, a novel, more-specific biomarker panel is still required, particularly for the early detection and prediction of complications. In addition, CVDs share common genetic biomarkers with other chronic conditions such as obesity and diabetes, which are also considered risk factors for the development of CVDs. Therefore, in the present study, we used a global proteomics approach to identify a set of dysregulated proteins in PBMCs from patients with reported CVDs and their matched controls. Subsequently, we validated the gene expression of selected proteins and assessed their circulating levels in plasma. We employed an integrative approach—both PBMCs and plasma

**Table 5. P-trend for the effect of diabetes.**

| Parameter | Controls non-diabetic | Controls diabetic | Cases non-diabetic | Cases diabetic | P-trend |
|---|---|---|---|---|---|
| | (n = 90) | (n = 62) | (n = 82) | (n = 126) | |
| *Anthropometric & Health Status* | | | | | |
| Age, years | 48.1±11.08 | 58.37±7.95 | 47.38±11.08 | 59.05±9.72 [&] | <**0.0001** |
| Sex (M/F) | 45/45 | 40/22 | 42/40 | 90/36 | **0.0032** |
| Weight, kg | 86.16±18.90 | 84.66±15.99 | 84.05±15.10 | 86.96±18.57 | 0.7847 |
| Waist, cm | 100.39±13.49 | 103.91±11.37 | 99.23±10.84 | 105.92±14.89 [&&&] | **0.0012** |
| Hip, cm | 110.81±12.13 | 108.69±11.50 | 108.92±10.64 | 109.61±12.89 | 0.6745 |
| BMI (kg/m2) | 31.45±5.99 | 31.06±5.11 | 31.02±5.00 | 31.64±6.51 | 0.8616 |
| SBP (mmHg) | 128.69±17.60 | 143.95±19.67 | 131.29±19.79 | 140.50±19.53 [&&] | <**0.0001** |
| DBP (mmHg) | 79.24±11.82 | 81.32±11.74 | 78.99±11.60 | 77.07±12.31 [#] | 0.1371 |
| Smoking (%) | 34.44 | 22.58 | 32.93 | 19.05 | **0.0336** |
| *Circulating Markers* | | | | | |
| FBG (mmol/l) | 5.03±0.5 | 8.39±3.06 | 5.13±0.56 | 8.78±3.80 [&&&] | <**0.0001** |
| Hb1Ac | 5.61±0.48 | 8.27±1.79 | 5.65±0.47 | 8.36±2.09 [&&&] | <**0.0001** |
| TC (mmol/l) | 5.41±0.93 | 5.13±1.15 | 5.07±1.02 [*] | 4.74±1.43 [#&] | **0.0008** |
| TG (mmol/l) | 1.50±0.81 | 1.92±1.18 | 1.52±0.85 | 1.92±1.68 [&] | **0.0208** |
| HDL (mmol/l) | 1.23±0.39 | 1.07±0.32 | 1.16±0.41 | 1.06±0.30 [&&] | **0.0023** |
| LDL (mmol/l) | 3.49±0.79 | 3.19±0.98 | 3.22±0.87 [*] | 2.80±1.14 [#] | <**0.0001** |
| hsCRP (µg/l) | 2.18 (0.20–11.9) | 2.62 (0.36–13.7) | 3.16 (0.44–10.50) [*] | 3.05 (0.09–11.85) | 0.4987 |
| oxLDL (µg/l) | 22.15 (1.32–69.78) | 16.04 (3.44–57.35) | 17.82 (2.58–55.71) [*] | 15.16 (0.03–61.21) | **0.0012** |
| MMP9 | 96.16 (8.23–398.63) | 90.58 (0.20–438.23) | 74.67 (5.46–998.14) | 85.50 (0.51–808.82) | 0.9952 |
| PR3 | 62.05 (14.05–571.30) | 61.20 (17.20–248.65) | 91.03 (5.35–514.60) [**] | 81.05 (16.30–705.45) [##] | **0.0254** |
| Defensin | 6.03 (5.71–83.62) | 6.03 (5.64–21.89) | 5.98 (5.68–12.76) | 6.02 (5.63–115.65) | 0.5038 |
| Annexin A3 | 2.09 (0.56–11.39) | 1.88 (0.68–21.45) | 2.24 (0.66–6.14) | 2.10 (0.73–8.39) | 0.9412 |

[*]: Control non-diabetic VS Cases non-diabetic

[#]: Control diabetic VS Cases diabetic

[&]: Cases non-diabetic VS Case diabetic

[*], [&] and [#]<0.05, [**], [&&] and [##]<0.01, [&&&]<0.001

from patients with CVD in direct crosstalk with the CVD processes—to facilitate the validation of newly discovered biomarkers in clinical samples from patients and, in turn, accelerate the translation of discovery into potential clinical application.

In the present study, we identified 47 dysregulated proteins with at least twofold increases or decreases between the two groups. Notably, dysregulated proteins are part of interrelated pathways involved in the progression of CVDs, such as tissue remodeling, chronic inflammation, and metabolic stress. Among the proteins identified in patients with CVD without diabetes, we selected four downregulated genes with concordant data both at the proteome and transcript levels (*MMP9, PR3, ANX3,* and *DEFA1*) for further validation in the plasma from a larger cohort including patients with CVD both with and without diabetes. Our data demonstrated that CVD influenced the protein levels in the cells (PBMCs) and the plasma differentially. Indeed, and despite the decreased levels of PR3, DEFA1, MMP9 and ANX3 (both protein and mRNA) in the PBMCs, PR3 circulating levels increased significantly in CVD cases, although no significant changes were observed in *DEFA1, MMP9,* and *ANX3* levels in the plasma. Although we did not validate the full list of the differentially expressed genes and proteins, here, we report the full list of potential markers, which could be investigated further as individual markers or as panels (Tables 1 and 2). Notably, most of the proteins are involved

**Table 6. P-trend for the effect of statin treatment.**

| Parameter | Controls non-Statins | Controls Statins | Cases non-Statins | Cases Statins | P-trend |
|---|---|---|---|---|---|
| | (n = 112) | (n = 40) | (n = 96) | (n = 111) | |
| *Anthropometric & Health Status* | | | | | |
| Age, years | 49.62±10.52 | 59.78±9.24 | 49.63±12.06 | 58.77±9.58 [&&&] | **<0.0001** |
| Sex (M/F) | 66/44 | 19/21 | 53/43 | 79/32 [## &] | **0.0249** |
| Weight, kg | 86.43±18.16 | 83.09±16.11 | 85.41±18.40 | 86.20±16.47 | 0.7535 |
| Waist, cm | 101.09±13.36 | 103.91±10.74 | 101.44±14.23 | 104.88±13.35 | 0.1289 |
| Hip, cm | 109.48±11.72 | 111.24±12.41 | 109.51±12.35 | 109.14±11.84 | 0.8212 |
| BMI (kg/m2) | 31.06±5.53 | 31.94±5.93 | 31.30±5.98 | 31.44±5.98 | 0.8649 |
| SBP (mmHg) | 135.03±18.92 | 134.60±22.63 | 136.79±22.51 | 136.90±17.96 | 0.8426 |
| DBP (mmHg) | 81.08±12.11 | 71.33±10.50 | 79.71±13.28 | 76.05±10.57 [&] | **0.0106** |
| Smoking (%) | 31.25 | 25.00 | 28.13 | 21.62 | 0.4230 |
| *Circulating Markers* | | | | | |
| FBG (mmol/l) | 6.08±2.38 | 7.30±2.93 | 6.74±3.78 | 7.87±3.12 [&] | **0.0002** |
| Hb1Ac | 6.43±1.68 | 7.43±1.85 | 6.84±2.40 | 7.70±1.76 [&&&] | **<0.0001** |
| TC (mmol/l) | 5.44±0.99 | 4.89±1.05 | 5.21±1.29 | 4.58±1.22 [&&&] | **<0.0001** |
| TG (mmol/l) | 1.67±1.05 | 1.66±0.84 | 1.64±1.29 | 1.86±1.53 | 0.5604 |
| HDL (mmol/l) | 1.15±0.35 | 1.20±0.44 | 1.13±0.40 | 1.07±0.31 | 0.1457 |
| LDL (mmol/l) | 3.54±0.80 | 2.92±0.94 | 3.33±0.95 | 2.65±1.05 [&&&] | **<0.0001** |
| hsCRP (µg/l) | 2.31 (0.20–13.7) | 2.5 (0.36–10.5) | 3.37 (0.32–10.5) | 2.91 (0.09–11.9) | 0.8266 |
| oxLDL (µg/l) | 18.45 (3.63–60.25) | 19.85 (1.32–69.78) | 18.97 (3.71–61.211) | 15.37 (0.03–57.67) | 0.1053 |
| MMP9 | 90.61 (8.23–438.23) | 98.95 (0.20–398.63) | 71.78 (3.82–998.14) | 87.48 (0.51–785.29) | 0.9921 |
| PR3 | 62.90 (19.05–571.30) | 59.23 (14.05–338.70) | 74.80 (27.25–514.60) [*] | 85.60 (5.35–705.45) [##] | 0.1073 |
| Defensin | 6.00 (5.63–115.65) | 6.44 (5.68–12.76) | 6.02 (5.64–60.18) | 6.04 (5.71–83.62) | 0.8135 |
| Annexin A3 | 2.24 (0.73–8.39) | 2.03 (0.66–6.14) | 1.99 (0.66–21.45) | 2.04 (0.56–11.39) | 0.8206 |

[*]: Control non-statins VS Cases non-statins

[#]: Control statins VS Cases statins

[&]: Cases non- statins VS Case statins

[*] and [&]<0.05, [##]<0.01, [&&&]<0.001

in biological processes associated with CVDs, such as exocytosis, neutrophil degranulation, vesicle-mediated transport, leukocyte activation, and response to stress.

MMP9 proteins belong to a family of metalloproteases that degrade extracellular matrix (ECM) and are involved in normal tissue remodeling; however, their induction is associated with several pathological conditions including chronic inflammation [14]. In humans but not rodents, neutrophil MMP9 is covalently linked with lipocalin and hence, protected from proteolysis while in various pathologies MMP9 are localized in the nucleus [15]. MMP9 proteins are also implicated in several stages of atherosclerosis involving leukocyte adhesion, cell migration, and matrix degradation [16]. Studies have reported elevated levels of MMP9 mainly in patients and animals with acute myocardial infarction (AMI) and acute coronary syndrome (ACS) [17–19]. DEFA1 is a member of the Defensin neutrophil peptides family, known to be cysteine-rich positively charged, that are secreted into circulation [20]. It was also reported to be stored in granules [21]. It binds to endothelial cells in vitro and accumulates in the intima of atherosclerotic vessels [20]. Recently, DEFA1 expression levels have been reported to be associated with coronary heart disease (CHD) in hyperlipidemic patients [22]. ANX3, a member of the calcium-dependent phospholipid-binding protein family, plays a role in the regulation of cellular growth and in signal transduction pathways [23]. It is also associated with cytoplasmic

**Table 7. Multivariate analysis for CVD prediction.**

| Model | dependent variable | Markers associated | Estimate | 95% Confidence Limits | P-value |
|---|---|---|---|---|---|
| 1 | *Annexin A3* | Diabetes | 1.665 | 0.924–3.001 | 0.0898 |
| | | TC | 2.404 | 1.395–4.141 | **0.0016** |
| | | Annexin A3 | 0.853 | 0.560–1.299 | 0.4596 |
| 2 | *Defensin* | Diabetes | 1.864 | 1.090–3.188 | **0.0230** |
| | | TC | 2.713 | 1.621–4.541 | **0.0001** |
| | | Defensin | 1.273 | 0.620–2.612 | 0.5109 |
| 3 | *MMP9* | Diabetes | 1.656 | 0.959–2.860 | 0.0702 |
| | | TC | 2.451 | 1.438 4.178 | **0.0010** |
| | | MMP9 | 0.988 | 0.783 1.246 | 0.9155 |
| 4 | *PR3* | Diabetes | 1.949 | 1.115–3.405 | **0.0192** |
| | | TC | 2.625 | 1.540–4.475 | **0.0004** |
| | | PR3 | 1.683 | 1.211–2.339 | **0.0019** |
| 5 | *oxLDL* | Diabetes | 1.766 | 1.027–3.037 | **0.0399** |
| | | TC | 2.758 | 1.643 4.628 | **0.0001** |
| | | oxLDL | 0.773 | 0.536–1.115 | 0.1684 |
| 6 | *hsCRP* | Diabetes | 1.690 | 0.956–2.990 | 0.0712 |
| | | TC | 3.053 | 1.753–5.314 | **<0.0001** |
| | | hsCRP | 1.369 | 1.039–1.804 | **0.0254** |
| 7 | *Annexin A3, Defensin, MMP9, PR3, oxLDL, hsCRP* | TC | 2.301 | 1.343–3.943 | **0.0024** |
| | | PR3 | 1.683 | 1.156–2.450 | **0.0066** |

Model 1–6: adjusted for age, sex, BMI, diabetes, TC, hypertension

Model 7: adjusted for age, sex, BMI, diabetes, TC, hypertension, hsCRP, PR3, oxLDL, Defensin, Annexin A3

granules and translocates to the plasma membrane in activated blood cells [24]. ANX3 expression increases in post-ischemic brains [25].

On the other hand, PR3 is a neutrophil serine protease, mainly stored in intracellular granules, that degrades ECM [26]. PR3 is also expressed on endothelial cells [27] and was reported to promote inflammatory response, induce vascular damage, and trigger endothelial cell apoptosis, particularly in Chronic Obstructive Pulmonary Disease (COPD) [28]. Notably, in the context of CVDs, PR3 is primarily reported to have deleterious effects in the pathogenesis of vascular inflammation such as vasculitis in Wegener's granulomatosis, and potentially in the prognosis for patients post-AMI [29]. However, a significant role of PR3 in disease development has emerged recently not only in COPD but also in other chronic inflammatory conditions, where PR3 is considered not only as an autoantigen but also for its involvement in the modulation of inflammatory pathways and cellular signaling [28].

The diverse functional and cellular roles of the genes and their expression products (RNA and proteins) and their expression profiles and associations with CVDs and other diseases seem to be context-dependent based on patient status, disease progression, and type of sample analyzed. For instance, the results of our proteomic screening and the RNA expression levels of the four genes confirmed a significant decrease in the markers in the PBMCs. MMP9 and ANX3 have been reported to be downregulated in subjects with coronary artery disease (CAD) with stable plaque without AMI or ACS compared with control subjects [30]. Numerous large studies on stable angiographically documented patients with CAD have failed to demonstrate any association between MMP9 and CAD, suggesting a downregulation of the enzyme [31, 32]. Similarly, DEFA1 expression was significantly higher and was associated with severe and AMI compared with patients with and without stable CAD [33–35]. Therefore, the

dysregulation of such protein levels seems to be associated more with acute CVD phases rather than a stable status phase. It is critical to note that our study patients did not report any recent CVD-related events and had stable statuses in addition to being treated with standard drugs, which may explain the decreasing trends of the proteins in the PBMCs.

Statins inhibit the secretion of MMP9 in smooth muscle cells and macrophages [36] and the expression [37] or the activity of PR3 [38, 39]. Nevertheless, we can rule out the possibility that the observed decrease in expression of the genes among cases was due to statin treatment, since there were no differences in the levels of the respective circulating proteins when the subjects with CVD were analyzed based on treatment or non-treatment with statin (Table 6). Interestingly, Kini *et al.* recently reported an enriched status of such gene transcripts, among others, in PBMCs from patients receiving high-dose statin therapies [40]. In addition, in the present PBMC transcriptome study, *PR3*, *DEFA1*, *MMP9*, and *ANX3* were clustered, highlighting their crosstalk in matrix remodeling and changes, inflammation, and immune response cellular functions. Previous studies have shown that PR3 activates pro-MMP2 and pro-MMP9 directly [41]. Similarly, PR3 binds to DEFA1 and regulates its extracellular expression and maturation during inflammation [42, 43]. Consistent with the results of the above studies, we observed a positive correlation between PR3 and DEFA1 in subjects with CVD as well as between MMP9 and ANX3 and DEFA1 in subjects with both CVD and non-CVD. In addition, our network (pathways) analysis revealed a link between the genes and their mechanistic pathways. Nevertheless, the causality and mechanistic contribution of the proteins to atherosclerosis and CVDs progression are only elucidated partially. For instance, animal studies have suggested a protective role of MMP9 with regard to the atherogenic process [44].

DEFA1 has a beneficial role, as reflected by its reduction of LDL-cholesterol and its "molecular brake" function on macrophage-driven inflammation, which facilitates the resolution of inflammation with minimal damage to tissues [45, 46]. Lastly, ANX3 downregulation has been reported to alleviate myocardial impairment in an AMI rat model [47]. ANX3 loci were enriched with ECM genes in a recent network-based identification of regulators of coronary artery disease, highlighting the potential role of ANX3 in tissue remodeling [48]. Therefore, the increased expression of the proteins may promote healing following atherosclerotic plaque rupture, resulting in the retardation of plaque expansion and the resolution of inflammation. Consequently, following an acute coronary event, an increase in plasma MMP9 concentrations, for example, could be a consequence of the healing response rather than the initial plaque rupture.

In contrast to the decreased expression levels of the four genes and proteins in the PBMCs of a small set of subjects, no differences were observed in the DEFA1 and ANX3 plasma levels when analyzing a larger cohort of subjects in the present study. The result could be attributed to the apparent stable CVDs status in the patients examined in the present study and suggests that PBMCs do not contribute considerably to the circulating forms of the proteins. Indeed, DEFA1 levels were similar between patients with middle-stage CHF (class I-II) and healthy controls but were significantly higher among patients with CHF in advanced stages (class III-IV) [34]. In addition, in the present report, no differences were observed in DEFA1 levels among patients with and without known CVDs. Circulating levels of DEFA1 were; however, reported to be linked to the CAD severity [35]. With regard to ANX3, little is known about its level in plasma when compared with other members of annexin family such as ANX1 [49]. Similarly, we did not observe a significant change in MMP9 levels in patients with CVD. Increased circulating levels of MMP9, however, have been reported in patients with ACSs [50] and peripheral arterial disease [51]. The absence of any significant increases in MMP9 levels in our study population could be attributed to the fact that both groups were obese and potentially experienced chronic subclinical inflammations [52]. In addition to the potential contribution of comorbidities and drug therapies to the MMP9 levels in plasma, the sample

collection methodology (serum or plasma, syringe or vacutainer), in addition to the relationship between systemic levels of MMP9 and the local atherosclerotic sites, could influence the levels. In addition, the usefulness of circulating MMP9 usefulness as a CHD biomarker is still debatable as it is also associated with many chronic comorbidities and may simply reflect the chronic inflammatory process in CVD [53].

Of note and out of the tested markers, PR3 significantly increased in the plasma of patients with CVD and was independently associated with CVD regardless of diabetes or treatment status. PR3 and its autoantibodies, ANCA (antineutrophil cytoplasmic antibodies) have been characterized extensively in the pathogenesis of vasculitis and tissue damage of Wegener's granulomatosis. In addition, more recently, its role has been reported to extend to other common vascular diseases [29]. It is suggested that the release of PR3 after blood cell activation represents a key step in the mediation of vascular injury. For instance, PR3 could determine the potential of mortality and incidence of heart failure following AMI, independently from established conventional risk factors [29]. Interestingly, PR3 plasma levels have recently been associated with obesity-induced metabolic disorders [54] and demonstrated to activate cytokines and modulate immune responses [55]. Therefore, they might play an important role in the development of inflammation and the progression of atherosclerosis. Similarly, in the present study, we observed a significant concomitant increase in PR3 plasma levels with an increase in adiposity markers (BMI and hip) in subjects with CVD-related events.

PR3 seems to have an array of functions in inflammatory processes. For example, PR3 is suggested to directly link inflammation to type 2 diabetes through the downregulation of insulin-like growth factor-1/IGFBP3 [56]. In a mouse CVD model, PR3 was suggested to play a role by triggering early atherosclerosis by rise to cytokine maturation [57]. Our current findings of increased PR3 circulating levels in patients with CVD could reflect an effect of general inflammation in atherosclerosis. However, PR3 levels in our study were not correlated with hsCRP or classical cardiometabolic markers. In addition, the high levels of circulating PR3 were inversely correlated with their expression in PBMC. The circulating PR3 could have originated from neutrophils, which are more abundant in blood but are not part of the PBMC pool. This claim would require validation through transcriptomic studies on whole blood cells.

Overall, and despite our observation that PR3 is a potential predictor of CVD progression patients, the current study had limitations that deserve consideration. Considering the limited understanding of the role of both cellular and circulating PR3 forms in CVD, our conclusions remain suggestive and await further validation through further studies. In addition, the cross-sectional nature of the present study does not facilitate the determination of causality; that is, whether the increase in PR3 was a cause or a consequence of CVD. Also, the patient selection process could have introduced some confounding variables, such as environmental factors, which could influence gene expression, and some clinical data on patients were missing, including some full medical histories. In addition, the results of the present study might not be generalizable to other populations because our study included the Arab population living in Kuwait only. Further investigations are warranted to address the putative causal role of PR3 in CVDs through prospective larger cohort studies including subjects at risk for CVD and assessing not only the PR3 expression levels but also its activity and autoantibodies. Knockout or upregulated gene expression of PR3 in animal models would also elucidate the contribution of PR3 to CVD onset and progression.

## Conclusions

In conclusion, to the best of our knowledge, this is the first study assessing the association between PR3 and the other markers in individuals with CVD in an Arab population. In

addition, our study used a high-risk group from a region with a high rate of obesity and diabetes. Our data suggest that the fluctuation of PR3 (increase in circulation or decrease in PBMCs) reflects underlying residual CVD risks even in a treated population. Prospective studies are required to establish the role of PR3 in CVD progression. Lastly, a panel of biomarkers may be required to predict residual risk and long-term cardiovascular outcome with greater accuracy.

## Supporting information

**S1 Table. Primer sequences used for real time PCR to analyse gene expression status of selected genes.**
(DOCX)

**S2 Table. Characteristics of subjects used in proteomics profiling and RT-PCR validation.**
(DOCX)

**S3 Table. List of protein groups identified with at least 2 unique peptides and in at least 4 out of the 8 LC-MS/MS runs using PBMCs collected from CVD patients and their matched controls.**
(XLSX)

## Acknowledgments

We are grateful to Clinical Laboratory and the Tissue Bank Core Facility at DDI for their contribution and Enago (www.enago.com) for the English language review.

Supplementary Materials: All used data and results are included in the manuscript and supplementary tables are included.

## Author Contributions

**Conceptualization:** Abdullah Bennakhi, Ali Tiss, Naser Elkum.

**Data curation:** Abdelkrim Khadir, Dhanya Madhu, Ali Tiss, Naser Elkum.

**Formal analysis:** Ali Tiss, Naser Elkum.

**Funding acquisition:** Monira Alarouj, Abdullah Bennakhi, Naser Elkum.

**Investigation:** Abdelkrim Khadir, Dhanya Madhu, Sina Kavalakatt, Preethi Cherian, Monira Alarouj, Jehad Abubaker, Ali Tiss, Naser Elkum.

**Methodology:** Abdelkrim Khadir, Dhanya Madhu, Sina Kavalakatt, Preethi Cherian, Abdullah Bennakhi, Jehad Abubaker, Ali Tiss, Naser Elkum.

**Project administration:** Monira Alarouj, Abdullah Bennakhi, Ali Tiss, Naser Elkum.

**Resources:** Monira Alarouj, Abdullah Bennakhi, Ali Tiss, Naser Elkum.

**Software:** Naser Elkum.

**Supervision:** Monira Alarouj, Jehad Abubaker, Ali Tiss, Naser Elkum.

**Validation:** Abdelkrim Khadir, Ali Tiss, Naser Elkum.

**Visualization:** Ali Tiss, Naser Elkum.

**Writing – original draft:** Abdelkrim Khadir, Ali Tiss.

**Writing – review & editing:** Abdelkrim Khadir, Dhanya Madhu, Sina Kavalakatt, Preethi Cherian, Monira Alarouj, Abdullah Bennakhi, Jehad Abubaker, Ali Tiss, Naser Elkum.

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
