## [Decision Letter · Decision Letter 0]

14 Nov 2019

PONE-D-19-27319

PR-3 levels are impaired in plasma and PBMCs from Arabs with cardiovascular diseases

PLOS ONE

Dear Dr. Tiss,

Thank you for submitting your manuscript to PLOS ONE. After careful consideration, we feel that it has merit but does not fully meet PLOS ONE’s publication criteria as it currently stands. Therefore, we invite you to submit a revised version of the manuscript that addresses the points raised during the review process.

Please pay careful attention to all the comments made by the reviewers and then proceed to address this in a point-by-point fashion. 

We would appreciate receiving your revised manuscript by 13 January 2020. To enhance the reproducibility of your results, we recommend that if applicable you deposit your laboratory protocols in protocols.io, where a protocol can be assigned its own identifier (DOI) such that it can be cited independently in the future. For instructions see: http://journals.plos.org/plosone/s/submission-guidelines#loc-laboratory-protocols

We look forward to receiving your revised manuscript.

Kind regards,

M. Faadiel Essop

Academic Editor

PLOS ONE

Journal Requirements:

  "The project (RA-2010-004) was approved by Ethical Review Committee at DDI. All subjects signed informed consent forms. ".

a.Please amend your current ethics statement to include the full name of the ethics committee/institutional review board(s) that approved your specific study.

b.Once you have amended this/these statement(s) in the Methods section of the manuscript, please add the same text to the “Ethics Statement” field of the submission form (via “Edit Submission”).

3. Your ethics statement must appear in the Methods section of your manuscript. If your ethics statement is written in any section besides the Methods, please move it to the Methods section and delete it from any other section. Please also ensure that your ethics statement is included in your manuscript, as the ethics section of your online submission will not be published alongside your manuscript.

5.  We noticed you have some minor occurrence of overlapping text with the following previous publication(s), which needs to be addressed:

https://www.hindawi.com/journals/dm/2018/9529621/

In your revision ensure you cite all your sources (including your own works), and quote or rephrase any duplicated text outside the methods section. Further consideration is dependent on these concerns being addressed.

6. Please refer to any post-hoc corrections to correct for multiple comparisons during your statistical analyses. if these were not performed please justify the reasons. Please refer to our guidelines for assistance (https://journals.plos.org/plosone/s/submission-guidelines.#loc-statistical-reporting).

Reviewers' comments:

Reviewer's Responses to Questions

**Comments to the Author**

1. Is the manuscript technically sound, and do the data support the conclusions?

Reviewer #1: Yes

Reviewer #2: Partly

Reviewer #3: Yes

2. Has the statistical analysis been performed appropriately and rigorously? 

Reviewer #1: I Don't Know

Reviewer #2: Yes

Reviewer #3: Yes

3. Have the authors made all data underlying the findings in their manuscript fully available?

Reviewer #1: Yes

Reviewer #2: Yes

Reviewer #3: Yes

4. Is the manuscript presented in an intelligible fashion and written in standard English?

Reviewer #1: Yes

Reviewer #2: Yes

Reviewer #3: Yes

5. Review Comments to the Author

Reviewer #1: In the present MS, Authors Abdelkrim Khadir and collaborators, investigated cardiovascular disease (CVD) biomarkers in circulating blood cells (PBMCs) and plasma from Arab obese subjects with and without CVD and diabetes.

By using proteomics tools Authors identified Proteinase-3 (PR3) as a putative independently biomarker associated with CVD in the plasma and PBMCs.

This work deals about a subject of great interest and arises from a very capable group and used valuable techniques.

That said I have few points that should be addressed by the Authors:

1 I have a little remark concerning the term "Arabs" in the title of the MS: "PR-3 levels are impaired in plasma and PBMCs from Arabs with cardiovascular diseases". I understood the study were performed on Human adults living in Kuwait. As inhabitants of Kuwait are called Kuwaitis, I thought the term "Kuwaitis" would be more appropriate. Please give me your opinion on this.

2 The abstract of the MS is not very clear-cut and therefore could be improved. To my point of view it lacks a logical progression. It should be more informative in terms of results.

Aims of the Authors is to identify CVD biomarkers in blood cells and plasma from Arab obese subjects with and without CVD and diabetes. Then, how come there are only two groups "Human adults with CVD (n = 208) and controls (n = 152)"?

Authors employed "a shotgun proteomic profiling approach on PBMCs isolated from a subset of the subjects, and differentially expressed proteins selected between the two groups were validated at the mRNA level using RT-PCR." What subset of subjects did Authors use? How many differentially expressed did the Authors identified? Why validation was made at the mRNA level and not the protein level?

Then after, in the abstract Authors wrote about Proteinase-3 (PR3), Annexin-A3 (ANX3), Defensin (DEFA1), and Matrix Metalloproteinase9 (MMP9), which plasma levels were measured by ELISA. What is the link with was written before and after this statement?

Just after, they wrote "We identified 47 dysregulated proteins", again I do understand the link and progression in the abstract.

In the abstract, sentence starting with "Despite the decreases..." is very long and complex. It should be rephrased and simplified.

3 introduction is very readable and clear. End of the introduction nicely itemize the different objectives of the work (pls refer to this when rephrased your abstract).

4 page 11, a reference should be included for the "Declaration of Helsinki" statement.

page 11 lane 99 " low-density Lipoprotein" no capital for lipoprotein

5 method section page 11, Authors should develop methods used for blood parameter measurements, TG LDL ...How did Authors measured oxidized LDL?

6 page 31 table 3. To my point of view, term "gender" should be replace by "sex" which is used in Science and corresponds to the genetic equipment (22 chr + XX or 22 ch +XY)

7 page 36 legend of figure 1 should be developed. What is the signification of the colors of the lines, their width ...

8 Page 14 lane 300, 302 and 303 Authors wrote CAD. I did not find the meaning of this acronym.

9 I think Authors should develop a bit more about biomarkers they identified (PR3, DEFA1, MMP9 and ANX3) in terms of tissue origin, protein localisation and role in the body.

10 Page 15 lane 341 Authors wrote “when analyzing a larger cohort of Arab ethnic groups (ref).”

11 Page 16 lane 358 Authors wrote “CHD biomarker CHD”

12 Authors included a limitation section at the end of the MS. If they write the present work is mostly descriptive, Authors could also develop on researches needed to get insight into a putative causal role of PR3 in CVD.

Reviewer #2: Review on the manuscript entitled: “PR-3 levels are impaired in plasma and PBMCs from Arabs with cardiovascular diseases” by Khadir A et al.

This article’s aim is to identify novel CVD markers in circulating blood cells and plasma from Arab obese subjects with and without CVD and diabetes. A number of biochemical parameters (PR3, ANX3, DEFA1 and MMP9) were analyzed using a shotgun proteomic profiling approach on PBMCs isolated from a subset of subjects.

This was quite an extensive study that was well designed and equally well written. That being said there are some aspects that the authors need to attend to.

• In the abstract it is not mentioned whether the study is cross-sectional or longitudinal, though this is mentioned later on in the discussion.

• Furthermore, the methodology should clarify how the sampling process was done, stating both the inclusion and exclusion criteria not only in the abstract but also for the manuscript.

• At the same time, the authors do mention that the individuals were “apparently” healthy. Who was determining the health status of the participants in the study? And this comes back to the sampling method and/or exclusion vs. inclusion criteria.

• It is stated both in the abstract and the introduction that the shotgun proteomic profiling techniques were used. However, the methods section does not mention shotgun profiling as such though it is described. It would be helpful to mention it as well.

• The discussion section was well executed and answered most of the questions I had, in particular the limitations of the study. One in particular on the different levels of PR3 in the circulation and that of the mRNA and protein expressed.

• There are some minor typos in the paper…

Line 93

Line 163

• Reference is missing in line 341

• The statements from line 354 and line 360 should be paraphrased for clarity.

Reviewer #3: The authors have taken a topic on cardio-metabolic diseases and using a shotgun proteomic approach, have shown novel proteins that are up and down-regulated in patients with CVD. This has been accompanied with their STRING network analysis which shows interlinks between the molecules and may present with proposed pathways. The identification of PR3, a neutrophil serine protease was up-regulated in CVD patients and presents itself as a novel marker in patients with CVD with diabetes and obesity.

6. PLOS authors have the option to publish the peer review history of their article (what does this mean?). If published, this will include your full peer review and any attached files.

Reviewer #1: No

Reviewer #2: No

Reviewer #3: No

---

## [Author Response · Author response to Decision Letter 0]

21 Nov 2019

Please see details in the attached document

---

## [Decision Letter · Decision Letter 1]

26 Dec 2019

PR3 levels are impaired in plasma and PBMCs from Arabs with cardiovascular diseases

PONE-D-19-27319R1

Dear Dr. Ali Tiss,

We are pleased to inform you that your manuscript has been judged scientifically suitable for publication and will be formally accepted for publication once it complies with all outstanding technical requirements.

With kind regards,

M. Faadiel Essop

Academic Editor

PLOS ONE

Additional Editor Comments (optional):

Reviewers' comments:

Reviewer's Responses to Questions

**Comments to the Author**

1. If the authors have adequately addressed your comments raised in a previous round of review and you feel that this manuscript is now acceptable for publication, you may indicate that here to bypass the “Comments to the Author” section, enter your conflict of interest statement in the “Confidential to Editor” section, and submit your "Accept" recommendation.

Reviewer #1: All comments have been addressed

Reviewer #2: All comments have been addressed

2. Is the manuscript technically sound, and do the data support the conclusions?

Reviewer #1: Yes

Reviewer #2: Yes

3. Has the statistical analysis been performed appropriately and rigorously? 

Reviewer #1: Yes

Reviewer #2: Yes

4. Have the authors made all data underlying the findings in their manuscript fully available?

Reviewer #1: Yes

Reviewer #2: Yes

5. Is the manuscript presented in an intelligible fashion and written in standard English?

Reviewer #1: Yes

Reviewer #2: Yes

6. Review Comments to the Author

Reviewer #1: In the present revised version of their MS, Authors Abdelkrim Khadir and collaborators, correctly addressed all points I raised. Hence I recommend publication.

Reviewer #2: (No Response)

7. PLOS authors have the option to publish the peer review history of their article (what does this mean?). If published, this will include your full peer review and any attached files.

Reviewer #1: No

Reviewer #2: No

---

## [Editor Report · Acceptance letter]

31 Dec 2019

PONE-D-19-27319R1 

PR3 levels are impaired in plasma and PBMCs from Arabs with cardiovascular diseases 

Dear Dr. Tiss:

I am pleased to inform you that your manuscript has been deemed suitable for publication in PLOS ONE. Congratulations! Your manuscript is now with our production department. 

With kind regards,

on behalf of

Dr. M. Faadiel Essop 

Academic Editor

PLOS ONE